# Extraction and Stabilization of Betalains from Beetroot (*Beta vulgaris*) Wastes Using Deep Eutectic Solvents

**DOI:** 10.3390/molecules26216342

**Published:** 2021-10-20

**Authors:** Omar A. Hernández-Aguirre, Claudia Muro, Evelyn Hernández-Acosta, Yolanda Alvarado, María del Carmen Díaz-Nava

**Affiliations:** División de Estudios de Posgrado e Investigación, Instituto Tecnológico de Toluca, Tecnológico Nacional de México, Tecnológico S/N, Colonia Agrícola Bellavista, Metepec 52149, Mexico; ohernandeza@toluca.tecnm.mx (O.A.H.-A.); ehernandeza2@toluca.tecnm.mx (E.H.-A.); yalvaradop@toluca.tecnm.mx (Y.A.); mdiazn@toluca.tecnm.mx (M.d.C.D.-N.)

**Keywords:** deep eutectic solvents, beetroot, wastes, betalain, pigments, extraction, degradation, stability, viscosity, kinetic model, food application

## Abstract

Deep eutectic solvents (DES) using magnesium chloride hexahydrate [MgCl_2_·6H_2_O] and urea [U] proportions (1:1) and (2:1), were prepared for their use as extracting and stabilizer agents for red and violet betalains from beetroot (*Beta vulgaris*) waste. The synthetized DES [MgCl_2_·6H_2_O] [U] showed similar properties to eutectic mixtures, such as, liquid phase, low melting points and conductivity, thermal stability, and variable viscosity. In turn, betalain DES extracts (2:1) exhibited compatibility in the extraction and recovery of betalains from beetroot wastes, showing a betalain content comparable to that of betalain extracts. Betalain stability was determined by degradation tests; the exposure conditions were visible light (12 h), molecular oxygen from atmospheric air and environmental temperature (20–27 °C) for 40 days. The kinetic curves of the betalain degradation of water samples depicted a first-order model, indicating the alteration of a violet colouration of betalains from beetroot waste for 5–7 days. However, betalains from DES extracts were kept under visible light for 150 days, and for 340 days in storage (amber vessels), achieving a stability of 75% in comparison with initial beet extracts.

## 1. Introduction

The industrial processing of fruits and vegetables produces waste that is generally disposed of as organic trash, leading to environmental pollution. However, this waste is an important source of phytochemicals and natural pigments that can be recovered and used to develop functional foods, pharmaceutical products, and other valuable compounds for the industry [1,2].

Specifically, beetroot waste contains a great quantity of betalains because they are found throughout the root. This waste includes peel and pulp, and is produced by the industrial manufacture of juice, jam, drinks, and sugar beet. Large amounts of waste with a high content of betalains are also obtained for ethanol fermentation by beetroot extracts.

Betalains are readily available biomolecules used as food colorants; they are known as “beetroot red”, covering a gamma of red pigments. Betalains are manufactured as powders (by freeze or spray drying) or extracts (by vacuum concentration of beet juice to 60–65% total solids), containing from 0.3% to 1% of pigment [3]. Nevertheless, the commercialized betalain pigments are highly susceptible to temperature and pH (T > 60 °C, pH > 5), because the aqueous content affects their chemical structure, causing rapid degradation and colour loss [4].

The current situation regarding betalains has led to a plethora of research regarding their stabilizing and colour preservation applications [5]. The inactivation of deleterious enzymes, and the addition of antioxidants and/or chelating additives have been suggested as ways of retaining the colouring strength of betalains [6]. However, despite these solutions, the short life of betalains continues to be a problem for many applications in the food industry. Thus, more research is necessary to extend pigment stabilization and retain the original colouration.

Recently, extractant substances known as deep eutectic solvents (DES) and natural deep eutectic solvents (NADES), have been used as alternative subtraction agents for isolating biomolecules [7,8]. Their use have numerous environmental advantages; they are ecologically friendly [9] and have a bright future in terms of their application in separating and recovering high-value products in process industries [10], mostly for the isolation and recovery of functional molecules (terpenoids, carotenoids, and flavonoids) from plants, fruit, and vegetables, with optimal results [11,12]. Mainly, extraction techniques and the optimization of process conditions by DES, including an economic analysis of DES usage as extractive agents are available [13]. However, the research on this topic is recent; therefore, there are still many aspects to resolve. The recovery of extracted compounds and DES solvents as well as DES recyclability for their reuse are the most important challenges [13,14].

More information regarding this topic is available in recent articles such as ones by Ameer et al. [15], Cunha and Fernandes [16], Vanda et al. [17], and Renard [18].

The green solvents, DES and NADES, are eutectic mixtures with two or three components showing similar properties to ionic liquids, such as, low melting points compared to parent species, low vapor pressure, good ionic conductivity, thermal stability, modifiable viscosity, miscibility, and solubility. In addition, their controllable polarity allows one to increase the stability and selectivity of extracts and the extraction yield [5], which further increases their range of industrial applications.

In accordance with the extracting requirements of extracting, DES or NADES are obtained by mixtures of two or more solid organic or inorganic substances, forming a single liquid phase. DES have been habitually synthetized by type II or III mixtures of choline chloride (ChCl), acting as hydrogen bond acceptors (HBA), and mixtures of quaternary ammonium salt such as amide, carboxylic acid, or alcohol moiety, and ammine, acting as a hydrogen bond donator (HBD). However, NADES are synthetized by natural HBD as sugars and organic acids, and HBA is equal to DES [6,7,14]; due to their similar structures, DES and NADES are usually identified as DES, regardless of their nature [8,9].

Currently, DES are also being analysed in terms of biomolecular extraction for pigment use [5,14,19,20]. However, there is a small number of studies focused on this line of enquiry. Particularly, the use of DES as extractant agents of betalains has not been reported either.

To test the effect of DES on the extraction and stability of betalains, the present research analysed a different method of betalain extraction from beetroot waste, using eutectic mixtures. The DES were prepared with urea (HBD) and magnesium chloride (HBA) in two molar proportions (1:1 and 2:1). DES were characterized according to their structure, viscosity, conductivity, and thermal behaviour. The stability of the betalain extracts was measured according to changes in their colouration and structure. The obtained results will contribute to the knowledge of DES behaviour and their application in betalain extraction, stabilization, and use as pigmented biomolecules.

## 2. Results and Discussion

### 2.1. Chemical Structure of DES

Chemical structures of DES [MgCl_2_·6H_2_O][U] (1:1) and (2:1) were analogous, presenting similar FTIR spectres. Figure 1 shows the FTIR spectra of DES 1:1, including the standard spectra of DES components, [U] and [MgCl_2_·6H_2_O].

Individual spectra of [MgCl_2_·6H_2_O] presented intense sharp bands at 1604 cm^−1^ which were attributed to the bending vibrations of the H atoms in water ligand with Mg^2+^. The bands at 3309 and 3197 cm^−1^ corresponded to the characteristic absorption peak of water molecules and the hydrogen bond of Cl–H in the MgCl_2_. A similar image of [MgCl_2_·6H_2_O] was found in Wang et al. [21]. In turn, spectra of [U] showed bands at 3340 and 3430 cm^−1^, which were linked to –NH_2_ groups. Additionally, stretching vibrations at 1677 and 1461 cm^−1^ from C–O and C–N bonds were observed, respectively, while a stretching vibration at 1571 cm^−1^ was characteristic of the carbonyl group (–C=O).

The FTIR of obtained DES [MgCl_2_·6H_2_O] [U] (1:1) confirmed the eutectic mixture, showing characteristic bands of Cl–H hydrogen bonds at 3322–3209 cm^−1^. However, a widening of the band at 3430 cm^−1^ was also observable due to the abundant hydrogen bonds in the DES [18]. The presence of Cl^−^ from MgCl_2_ was observed in the form of a band at 590 cm^−1^; whereas the –NH_2_ group from [U] in the DES was also detected by an enlargement (stretching) of the band at 1610 cm^−1^, manifesting an increase in the interactions of the hydrogen from –NH_2_ with Cl^−^. In addition, the absence of OH^−^ groups from water and the formation of hydrogen bonds between the Cl–H led to a pronunciation of the Cl^−^ group.

The interaction between the hydrogen bond from [U], which was used as a donor (HBD), and the chlorides Cl^−^ from the halogen salt [MgCl_2_·6H_2_O], which were used as acceptors (HBA), allowed the melting point of both salts to decrease [22]. This was evidence of the DES formation, which was confirmed in the following Sections.

### 2.2. Viscosity Behaviour of DES

Figure 2 shows the viscosity behaviour of DES [MgCl_2_·6H_2_O][U] (1:1) and (2:1) as a function of shear stress (0–160 Pa) (left) and temperature (0–95 °C) (right). Both molar proportions of DES exhibited a solution with a viscous appearance; however, they presented a different viscosity behaviour.

DES (1:1) achieved the highest viscosity value at 25 °C (0.535 Pa.s), whereas DES 2:1 presented a slightly lower value of 0.4623 Pa.s.

According to the fluid type, DES (1:1) and (2:1) showed non-Newtonian behaviour (range 0–60 Pa); specifically, as pseudoplastic and dilatant fluids, respectively, whereas the range >60 Pa displayed similar Newtonian behaviour for both proportions.

Regarding the viscosity variation as a function of temperature, DES (1:1) and (2:1) exhibited a reduction in viscosity with an increasing temperature, which was caused by weak molecular interactions.

DES (1:1) showed most viscosity in the range of 0–30 °C. After the last range (>30 °C), both proportions of DES presented an equivalent viscosity and the same behaviour.

The viscosity difference in DES was associated with the proportion of the anion [MgCl_2_·6H_2_O] present. A particularly high concentration of this ion in DES (2:1) inhibited the capacity to form several hydrogen bonds and the availability for acceptance by the donor group [U], leading to less viscosity.

The DES proportion 1:1 indicated the most extensive H-bonding network between [U] and [MgCl_2_·6H_2_O], leading to a slightly higher viscosity value.

Similar viscosity values in the >30 °C range were attributed to a comparable water content in both proportions of DES. The increase in viscosity of DES compared to water was attributed to enhanced van der Waals forces relative to the hydrogen bonding [23]. In addition, the DES viscosity was not comparable with individual components because the holes for the solvents or ions to move into are of different sizes, and the size of the ions themselves is different [24].

### 2.3. Physicochemical Characteristics of DES

The physicochemical characteristics of prepared DES [MgCl_2_·6H_2_O][U] (1:1) evidenced the formed DES as eutectic mixtures type IV with a general formula nMgCl_2_ + U. However, they were different in appearance. DES (2:1) was a viscous, homogeneous, colourless liquid, whereas DES (1:1) was a whitish, more viscous liquid.

Table 1 presents the physicochemical properties of DES (1:1) and (2:1), showing that the rate and hydration of magnesium chloride played an important role in obtaining DES and their properties. This is explained below.

The melting points of both DES were lower than the pure components, [U] (135 °C ± 1) and [MgCl_2_·6H_2_O] (118 °C ± 0.5), demonstrating the DES formation, because systems with an extreme reduction in the melting point can be recognized as DES [25]. However, herein was observed that the (2:1) proportion was slightly higher than the (1:1) proportion, indicating the influence of the anion proportion [MgCl_2_·6H_2_O] on the melting points of these DES.

The decrease in melting points in DES has been explained in different ways. According to Sun et al. [26], coulombic interactions between the cation and the anion in DES cause the delocalization of the negative charges through hydrogen bonding, leading to reduced melting points. Consequently, a higher amount of [MgCl_2_·6H_2_O] in DES could reduce the number of coulombic interactions and increase the melting point of DES (2:1). However, Ashworth et al. [27] revealed that a wide variety of bond types (neutral, ionic, doubly ionic) are present in DES, showing competitive HBA-anion and HBD-cation interactions, which may well promote the reduction in the melting point of DES.

Nevertheless, recent studies of [ChCl][U] by neutron diffraction experiments and simulated tests indicated that charge delocalization is not responsible for lowering the melting point [12]. Consequently, the reduction in the secondary electrostatic cation–anion interactions, and an overall decrease in anion coordination at the cation charge centres could cause changes in the melting point of DES [ChCl][U], but the HOH–Cl^−^ hydrogen bond was not a key interaction in the formation of this DES, because this structure remained unperturbed in the solid salt lattice.

Furthermore, the density of DES [MgCl_2_·6H_2_O][U] showed imperceptible differences. DES (2:1) was slightly higher than (1:1), because the density decreased when the percentage of salt increased in the DES; however, the increase in (2:1) was almost imperceptible, because the density decreased with the water percentage in the DES (2:1). The analogous carbon number of cations of [U] in both proportions also allowed a little variation.

In comparison with the pure-compound vacancies, DES presented an increment in the density, which was explained because the number of vacancies in the molecular structure of DES had increased.

The densities of DES [MgCl_2_·6H_2_O][U] (1:1) and (2:1) were comparable to ionic liquids based on [ChCl] (1:1) with [U] (1.25 g/mL), oxalic acid, triethanolamine and ethylene glycol (1.06–1.2 g/mL) [28], methylimidazolium cations, hexafluorophosphate anions [C_4_mim][PF_6_], [C_8_mim][PF_6_], and they were similar to water (1.2–1.35 g/mL) [26].

Electrical conductivity from DES presented reduced values compared to individual components. Sequentially, the highest conductivity of DES was observed in proportion (2:1). The difference between both DES proportions was attributed to the high viscosity of DES (1:1), which was observed in Section 3.2. A high viscosity causes the ionic species not to be completely dissociated in the mixture and, thus, their movement is not independent [23]. Therefore, coulombic interactions between cations and anions were reduced. Contrarywise, the increase in anion [MgCl_2_·6H_2_O] in DES proportion (2:1) reduced the viscosity of DES and, thus, coulombic interactions increased its electrical conductivity.

Reports regarding the preparation and characterization of DES [MgCl_2_·6H_2_O][U] have not been described in the scientific literature; however, Wang et al. [21] used [MgCl_2_·6H_2_O] and [ChCl] in the (1:1) proportion for DES preparation. Physicochemical properties showed that the melting point was found to be 16 °C; furthermore, a high conductivity and a low viscosity was found in this DES. However, the results did not coincide with those of [MgCl_2_·6H_2_O] [U]. The report was used to indicate the difference, which was attributed to the HBA and HBD arrangement. In addition, the properties of other prepared DES such as [ChCl][A], [ChCl][U], and [Emim][MeSO_4_] also exhibited modifications in the density and conductivity values, because their properties were generally different in comparison with individual components.

In terms of thermal behaviour, both DES [MgCl_2_·6H_2_O][U] displayed a high thermal stability compared to individual components (Ts > 170 °C). The degradation temperature (Td) was found at 210 and 215 °C for proportions (2:1) and (1:1), respectively (maintaining a weight > 5%). The discrepancy in Td proportions demonstrated that the mixtures presented different structures. Thus, Td was dependent on the intermolecular interaction and coordinating nature of the ion in the mixtures. In addition, the water content influenced Td, indicating the highest value for DES proportion (1:1). In sequence, the degradation of [U] was similar to the values reported by Chemat et al. [7].

The glass transition temperature (Tg) was detected at −45.8 and −40.1 °C for the (1:1) and (2:1) proportions, respectively; a constant increase in the endothermic process was observed, due to the relaxation of hydrogen bonds from DES [29,30,31]. The differences between DES proportions were attributed to the [MgCl_2_·6H_2_O] content in the mixture.

The values of Td for DES [MgCl_2_·6H_2_O][U] agreed with other synthetized DES such as [Tetrabutylammonium Cl][U] (4:1) and [ChCl][glucose] (1:1), showing a stability range of 124–200 °C for both DES [30,31]. Other reported data regarding DES, such as [Citric acid][ChCl], [Lactic acid][ChCl], [Oxalic acid][ChCl], and [Tartaric acid][ChCl], showed Td = 155, 197, 135 and 198 °C, respectively. However, information regarding DES [MgCl_2_·6H_2_O][U] was not found in previous reports; there are no data concerning their decomposition curves.

### 2.4. Beetroot Betalain Extraction by DES

Betalain extraction by DES [MgCl_2_·6H_2_O][U] at neutral and acidic pH were identified as BED7 and BED3, respectively, while betalain water extracts at a similar pH were described as BEW7 and BEW3.

Data were relative to betalain extraction by the DES [MgCl_2_·6H_2_O][U] (2:1) proportion, because the high viscosity of DES (1:1) affected the diffusion of betalains negatively, hindering their extraction at 20–25 °C (data of viscosity of proportion 1:1 were presented in Section 3.2). In this case, the anion of DES was a crucial factor for viscosity properties.

As a result, Figure 3 shows UV–Vis spectra from DES and BEW extracts, exhibiting similar bands; however, BEW samples presented absorption bands at 480 nm, which were identified as yellow betaxanthins, while bands at 535–540 nm were recognized as red-violet betacyanin. The difference between betaxanthin and betacyanin is that the substituent of betaxanthin is betalamic acid residue and the substituent of betacyanin is dihydroxyphenylalanine. The substituent of betaxanthins is commonly glutamine (vulgaxanthin) [5].

In turn, beetroot extracts, BED7 and BD3, presented comparable betacyanin bands; nevertheless, betaxanthin reduction was observed in their spectra. Furthermore, betacyanin presented an obvious absorption in the 270–280 nm UV range because of cyclo-3,4-dihydroxyphenylalanine (cyclo-Dopa) residue. The presence of cyclo-Dopa in betalains identified the betalamic acid (betalain chromophore), whose conjugation with cyclo-Dopa produced red–violet--coloured betacyanin; whereas, the conjugation with different amino acids or amines formed yellow betaxanthins [32].

UV–Vis spectra of BED3 extracts also showed the highest betalain content, due to the pH. The increase in the proton quantity favoured the hydrogen bond augmentation [33].

Table 2 presents data for the betalain content of beetroot extracts and the extraction yield (%). The order of Betacyanin content was observed as BED3 > BED7 > BEW3 > BEW7, whereas the total content of betalain was found in the 3.65–3.99 mg/g range in DES, and BEW extracts showed 3.49–3.55 mg/g.

The extraction yield indicated that DES solvent (2:1) exhibited compatibility in the extraction and recovery of betalains from beetroot waste, because its viscosity facilitated the suspension of beet betacyanin. In addition, acid extracts provided the highest yield of betalains; however, DES extracts exhibited a noticeable reduction in betaxanthins. This phenomenon was attributed to betacyanin stability and diffusivity in the extracts, as well as DES selectivity by betacyanin, trapping the molecule [34].

The results of the total betalain content from BEW and BED extracts were in accordance with [35], showing 3.80 mg/g on fresh weight. Through an HPLC analysis, it was found that the major constituents of red beet pigments were betalain and isobetalain. The results of Singh et al. [36] were also comparable. The authors achieved 2.42 and 4.59 mg/g of betalains, using microwave-assisted extraction and citric acid, as well as ethanolic water dissolutions as solvents, respectively.

The enhancement of betalain extraction with acidification was also previously corroborated by other authors [36,37]. They found that the increase in extraction occurs in a pH range between 3 and 5, using water as an extraction solvent.

### 2.5. Stabilization Analysis of Betalains in DES Extracts by Environmental Conditions of Light and Oxygen

DES and BEW extracts displayed initial purpura–red tonality from the beetroot; however, water extracts exhibited an alteration in the colouration and loss of pigment because environmental conditions of light and oxygen exhibited a negative impact on the betalain stability, increasing molecular reactivity during the oxidation of polyphenol oxidases [38].

Conversely, the highest colouration of betalains was well preserved in extracts BED7 and BED3 (2:1). A reduced quantity of water and oxygen contained in the samples extended the betalains’ life. The principal betalain reduction was detected after 27 days. However, after this time, the betalains were sustained, even with exposure to light.

Figure 4 exhibits the linear kinetic curves of betalain degradation from beet extracts BED3, BED7, BEW3, and BEW7. The kinetics confirmed a fast reduction in betalains in water extracts BEW7 and BEW3, as well as a colour stability in BED3 and BED7. The data here were relative to changes in betalain concentrations during the test, owing to the exposure of samples to environmental conditions for 40 days of visible light (12 h), and molecular oxygen from atmospheric air and environmental temperature (20–27 °C).

According to final colouration, a possible hydrolysis of the aldimine bond of betalains from water extracts resulted in the formation of betalamic acid (yellow colour) and the colourless cyclo-Dopa-O-ß-glucoside [38]. Hydrolytic reactions were linked to the interaction of oxygen with light, inducing the fast degradation of betaxanthins and betacyanin [5,6].

The probable dehydrogenation of phyllocactin and hylocerenin, forming other yellow-coloured degradation products, could also provide evidence of the degradation of betalains and the final colouration in water extracts [39]. In addition, the free sugars and nitrogenous compounds caused fermentation, probably contributing to the reduction in the violet colouration.

The pH also started the degradation of BEW3, because the increase in the interaction of H+ bonds with the protonation of the aromatic characteristic betacyanin group caused a rise in betalamic acid; consequently, a reduction in the pigment stability, and less colouration.

In turn, beet DES extracts BED7 and BED3 increased the stability of the betalains because the total betalain content was higher than water extracts, resulting in a 75% increase in beet water extracts. The red–violet colouration was also maintained during test degradation, exposing the condensation of betalamic acid to cyclo-Dopa (betacyanins) to obtain a maximum stability of betalains.

Furthermore, betalains from DES extracts were well preserved under visible light for 150 days, and 340 days in storage (amber vessels). Consequently, DES resulted selective in terms of betalains from beets, and the viscosity of (2:1) proportion was an additional factor to keep the betalain diffusion in the samples, because a high viscosity reduces the mass transfer of oxygen, preventing betalain degradation. Additionally, the probable conservation of amino compounds from betalains in DES extracts allowed the maintenance of the violet colouration because they are generally coloured.

According to results, Equation (1) describes the betalain degradation of DES extracts, expressing a first-order kinetic model. The general solution of Equation (1) is expressed in Equation (2), where Kb is the degradation rate constant of betalains, B_Ct_ is the betalains concentration in extracts at a definite time (day), BC_0_ is the betalain concentration at initial time zero, and t is the time of the test (0–40 day).
(1)$$\frac{-\mathrm{dBc}}{\mathrm{dt}}=K_{b}\mathrm{BC}$$
(2)$$\mathrm{Ln}\left(\frac{B_{c_{t}}}{{\mathrm{BC}}_{0}}\right)=K_{b}t$$

Table 3 exhibits the values of the K_b_ constant (line slope) and correlation coefficient (R^2^) by the linear regression of the linear kinetic model of betalain degradation from beetroot extracts.

The high values of R^2^ long confirmed a first-order reaction of betalain degradation from beetroot extracts by light and oxygen. In addition, the high values of K_b_ from extracts BEW3 and BEW7 validated the fast degradation of betalains in these extracts, while the K_b_ values from BED7 and BED3 indicated a slow degradation. Therefore, the presence of DES [MgCl_2_·6H_2_O][U] in extracts reduced the instability of betalains in the presence of light and oxygen.

Additionally, values from acid extracts BEW7 and BED7 presented higher K_b_ values than BED3 and BEW3, exhibiting more betalain degradation, which was attributed to the pH.

Similar behaviour was found in previous reports of betalain degradation by temperature [40]. In this test, a first-order kinetic model was also described, indicating an important degradation in the temperature range of 50–120 °C.

Other alternatives to control betalain stability have verified that encapsulation conserves their colouration. Particularly, Ravichandran et al. [41] and Antigo et al. [42] observed an increase in the stability of red beetroot pigments, and betalain microcapsules showed the lowest degradation constant and the longest half-life. However, betalain degradation was also observed after 7 days of storage at 30 °C.

## 3. Materials and Methods

The reagents used in the preparation of DES were analytical grade. Urea (NH_2_CONH_2_) was provided by Sigma-Aldrich (Sigma-Aldrich Química S de RL. de C.V., Toluca, México); the magnesium chloride (MgCl_2_·6H_2_O), HCl, and NaOH were provided by Fermont (Fermont Productos Químicos Monterrey, México).

The 5kg of beetroot (*Beta vulgaris*) waste used in this research was provided by a Mexican beetroot juice manufacturer (México city, México). The beetroot waste included peel and pulp from var. *red cloud*. The water content of the beetroot waste was determined as 550 ± 17 mg/g. The waste was distributed and stored in plastic bags of 100 g at 0 °C for later use. The loss of water from the thawed waste was found to be 533 ± 17 mg/g, indicating a 3.2% loss of water.

### 3.1. Preparation and Determination of DES Characteristics

DES were prepared by mixing magnesium chloride hexahydrate [MgCl_2_·6H_2_O] (HBA) and urea [U] (HBD), to give two molar proportions: (2:1) and (1:1).

The mixture was placed in a 500 mL closed flask for each molar ratio. The mixture was stirred at a controlled temperature between 50 and 60 °C until a homogeneous liquid was obtained.

The obtained products were identified as DES [MgCl_2_·6H_2_O][U] (2:1) and (1:1), according to the proportions. Characteristics such as density, viscosity, electrical conductivity, structure by functional groups, and thermal behaviour were determined for both DES. An Anton Paar RheolabQC (Graz. Austria) rotational viscosimeter was used to measure the density (mg/mL) and viscosity (Pa.s) of DES. Both parameters were determined in the temperature range of 0–90 °C and shear stress of 0–180 Pa. Electrical conductivity was measured at 25 °C using a Hanna conductivity meter (Rhode Island, USA). Functional groups were analysed and identified by Fourier Transform Infrared Spectroscopy (FTIR-ATR) Spectrum Varian 640-IR (Agilent, Santa Clara, CA, USA). The spectra conditions were 25 °C, 64 scans, 500–4000 cm^−1^ and 2 cm^−1^ resolutions, with respect to the appropriate background spectra. The thermal decomposition and thermodynamic transition properties of DES were determined by thermogravimetric analysis (TGA) and differential scanning calorimetry (DSC), using a PerkinElmer Simultaneous Thermal Analysis model 8000 (TA Instruments, New Castle, DE, USA), with a heating ramp of 10 °C for 60s, in a nitrogen atmosphere, at a flow rate of 0.7 mL for 60 s. Samples were heated from room temperature to 350 °C. The crucibles of aluminium with a 5 mm diameter were used in this test. Savitzky-Golay smoothing algorithm was employed for TGA curves.

### 3.2. Beet Betalain Extraction by DES

Betalains extraction from beetroot waste was carried out using the prepared DES [MgCl_2_·6H_2_O][U] (2:1) and (1:1) as extraction and stabilizing solvents. To measure the effect of pH on extraction efficiency, the DES were previously adjusted to pH 7 ± 0.3 and pH 3 ± 0.2.

Fresh beet pieces (0.5 g) were mixed with the DES solvent at a solid-to-liquid ratio of 1:30 g/mL (15 cm^3^) in a blender. After, ultrasonic assisted extraction of betalains was applied using an ultrasonic batch (Bransonic CPX1800H, Emerson, México) at 25 °C for 3 h with subsequent vortex agitation for 900 s. The liquid was separated from the beetroot mass by filtration, using Whatman filter paper No.1. The liquid was identified as DES extract from beetroot waste. Water extracts of beetroot waste were also obtained as blank; they were used for comparing DES extracts with conventional isolates. Water betalain extracts were obtained as DES extracts.

The extracts were characterized according to total betalain content (mg/g) and betalain yield (%). The methodology of Castellar et al. [43] and Equation (3) were used to calculate the total betalain content (mg/mL). Betalain colouration was measured in a PerkinElmer Lambda 25 UV–Vis (Waltham, MA, USA), considering betaxanthins and betacyanin as principal structures in betalain extracts.
(3)$$\mathrm{Total}\;\mathrm{betalain}\;\mathrm{content}\;(\mathrm{mg}/\mathrm{mL})=\frac{A\times \mathrm{DF}\times \;\mathrm{MW}\times 1000}{Ɛ\left(L\right)}$$
where A is the absorbance value of the extract at 535 nm for betacyanin and 480 nm for betaxanthin. DF is the dilution factor of the extract, and L is the path length of cuvette (cm). MW represents the molecular weight for betacyanin, 550 g.mol and 308 g.mol for betaxanthin. The extinction coefficients for betacyanin 60,000 M^−1^.cm^−1^ and 48,000 M^−1^.cm^−1^ for betaxanthin were relative to ε. A conversion factor of 1000 was used to convert g to mg.

Betalains yield was determined according to Equation (4), including betalain content in extracts from beetroot waste. According to DES solvent and pH (BED7, BED3), or water solvent and pH (BEW7, BEW3) and betalain content in extracts from whole beetroot (BEWW). BEWWW was determined as 4.67 mg/g.
(4)$$\mathrm{Betalains}\;\mathrm{yield}\;(\%)=\frac{\mathrm{BED}\;\mathrm{or}\;\mathrm{BEW}}{\mathrm{BEWW}}\times 100$$

### 3.3. Stabilization Analysis of Betalains in DES by Environmental Conditions of Light and Oxygen

Stabilization analysis of betalain extracts by DES and distilled water were studied by kinetic behaviour, monitoring the total betalain content at time zero (control) to 1, until 40 days of extract storage. The degradation was measured in exposed extracts under environmental conditions of visible light (12 h), molecular oxygen from air in the atmosphere and environmental temperature (25 °C). Betalain content was determined according to Equation (1), using the procedure indicated in Section 2.3. In addition, functional groups were also analysed by FTIR at time zero and after 30 days.

### 3.4. Statistical Analysis

All experiments and analytical determinations were conducted in triplicate. Data were processed with statistical methods in the OriginPro 2016 software 93.

## 4. Conclusions

In this work, betalains from beetroot wastes were extracted and stabilized using deep eutectic solvents (DES). DES were prepared using magnesium chloride hexahydrate [MgCl_2_·6H_2_O] as (HBD) and urea [U] as (HBA), both in proportions (1:1) and (2:1) to obtain eutectic mixtures of [MgCl_2_·6H_2_O][U] with adequate properties to extract, retain, and stabilize the betalains in the beetroot extracts.

In accordance with viscosity, DES (2:1) was used to extract betalains from beetroot waste, displaying a similar betalain content to water extracts. However, the kinetic curves of betalain degradation by exposing water samples to light and atmospheric oxygen described the alteration of the violet colouration from beetroot pigments. Betalain water extracts were degraded for 5–7 days, whereas betalains from DES extracts were well preserved under visible light for 150 days, and 340 days in storage (amber vessels), achieving a stability of 75% of the red violet colouration.

Based on the aforementioned data, DES [MgCl_2_·6H_2_O][U] were an effective extraction and stabilizing agent for betalain removal from beetroot wastes. This demonstrated betalain stability, and a suitable level of moisture, as well as miscibility with betalains.

The results obtained in the present study contribute to the field of DES research, and the novel separation techniques for application as natural pigments. However, the direct use of DES extracts is limited in food areas because the presence of DES affect’s organoleptic properties. Nevertheless, DES extracts could also be used as antifungal or antibacterial pigments. Therefore, more studies are necessary to suggest a global proposal of application. In addition, further research on the recovery and purification of betalains and DES recovery would provide more information to establish with certainty its usage in the food industry or other areas.

## Figures and Tables

**Figure 1 molecules-26-06342-f001:**
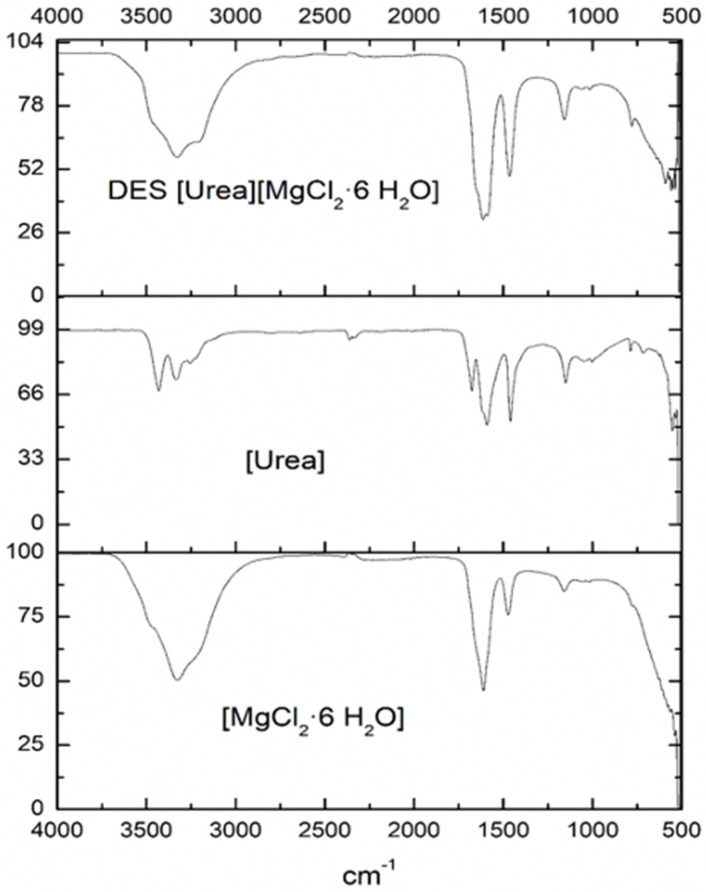
FTIR-ATR spectra of DES [MgCl_2_·6 H_2_O] [U] proportion (1:1).

**Figure 2 molecules-26-06342-f002:**
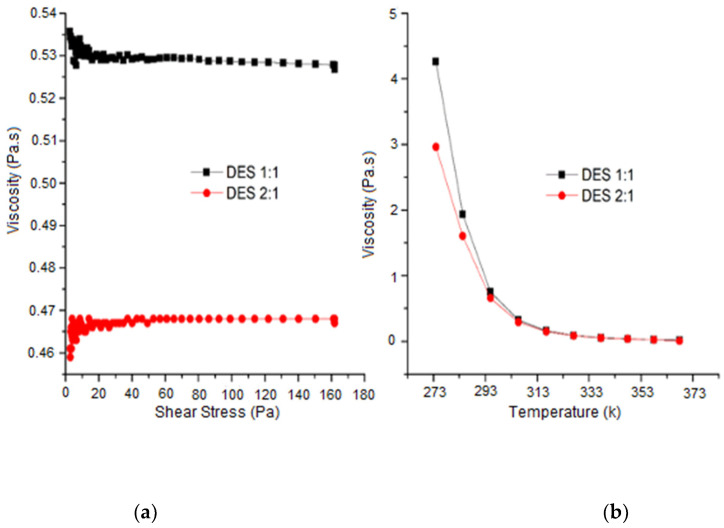
(**a**) Viscosity of DES [MgCl_2_·6H_2_O][U] as a function of shear stress (**b**) and temperature.

**Figure 3 molecules-26-06342-f003:**
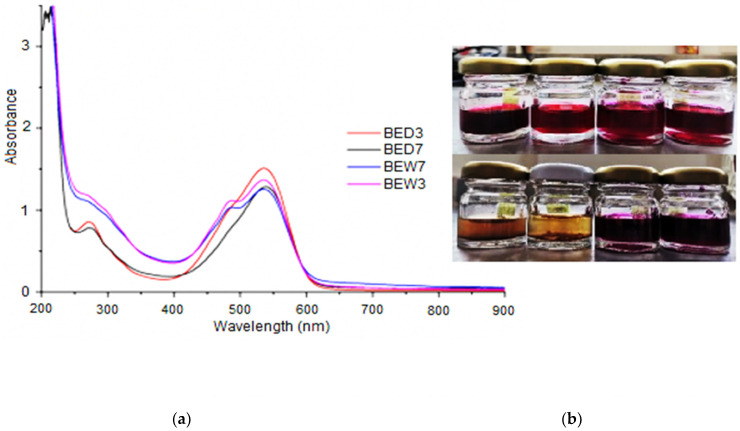
(**a**) UV–Vis spectra of extracts of beet by DES (2:1) pH 7 (BED7) and pH3 (BED3), and water extracts at similar pH (BEW3 and BEW7); (**b**) images of water and DES extracts from beetroot wastes (higher to lower), BW3, BW7, BED3, and BED7, at initial and final time of the test degradation of betalains (40 days).

**Figure 4 molecules-26-06342-f004:**
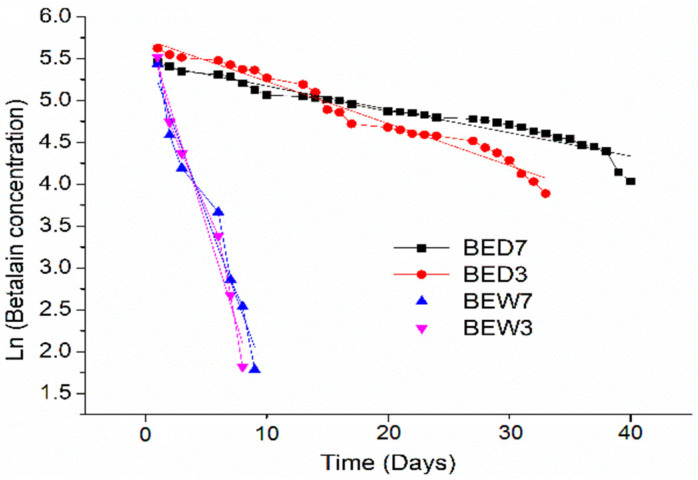
Linear kinetic of betalain degradation from beet DES (BED) and water extracts (BEW) for 40 days of exposure to visible light and environmental temperature (20–27 °C).

**Table 1 molecules-26-06342-t001:** Characteristics of prepared DES [MgCl_2_∙6H_2_O][U] proportions 1:1 and 2:1.

Physicochemical Properties	Individual Components of DES		DES Proportions	
	[U]	[MgCl_2_∙6H_2_O]	1:1	2:1
Water contained (mass fraction)	0	0.53	0.36	0.72
Melting point (°C)	135 ± 1	118 ± 0.5	19 ± 0.4	21 ± 0.3
Density (g/mL)	1.21 ± 0.5	1.27 ± 0.4	1.46 ± 0.3	1.47 ± 0.3
Electrical conductivity (mS/mL) 20 °C	5.2 ± 0.2	7.9 ± 0.1	3 ± 0.4	1.9 ± 0.4
Degradation temperature (Td/°C)	155 ± 2	197 ± 1	215 ± 1	210 ± 1
Glass transition temperature (Tg/°C)	-	-	−45.8 ± 1	−40.1 ± 1

**Table 2 molecules-26-06342-t002:** Betalain content on fresh weight and betalain yield (%) in beetroot DES (2:1) (BED7 and BED3) and water extracts (BW7 and BEW3).

Extraction Solvent	Betacyanin (mg/L)	Betaxanthin (mg/L)	Total Betalain (mg/g)	Betalain Yield (%)
BED7 (2:1)	234.23 ± 7.7	93.1 ± 8.5	3.65 ± 0.25	78.00 ± 11.
BED3 (2:1)	296.7 ± 7.6	98.5 ± 7.5	3.99 ± 0.26	83.75 ± 12
BEW7	229.9 ± 5.1	137.5± 4.8	3.49 ± 0.14	75.30 ± 6
BEW3	250.5 ± 8.8	148.4 ± 7.8	3.55 ± 0.20	77.31 ± 9

**Table 3 molecules-26-06342-t003:** Values of K_b_ constant and correlation coefficient R^2^ to the first-order kinetic model for betalain degradation from DES proportion 2:1 and water extracts.

Betalain Extracts	BED7	BED3	BEW7	BEW3
K_b_ (day^−1^)	0.0502	0.0279	0.4965	0.4725
R^2^	0.9694	0.9855	0.9779	0.9861

## Data Availability

Not applicable.

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
