# Peer review of "Extraction and Stabilization of Betalains from Beetroot (*Beta vulgaris*) Wastes Using Deep Eutectic Solvents"

_molecules, 2021, doi:10.3390/molecules26216342_

Round 1
Reviewer 1 Report
Present research by Hernandez-Aguirre et al. investigates extraction and stabilization of beetroot pigments. Deep eutectic solvents (DES) based on urea and magnesium chloride hexahydrate were synthetized, characterized and applied as extraction solvents. Betalain pigment yield was compared with traditional extraction with water and stability study was performed on obtained extracts. Extracts obtained by DES provided clear advantages comparing to conventional extracts and present study should be very interesting since it provides interesting concept of waste valorization. Following remarks and questions must be accessed by the authors in order to improve the paper prior the further reconsideration.
- Lines 30-32: Current application of beetroot was should be given.
- Lines 53-55: Please brief explanations of these references instead of just listing the authors.
- Line 60: the stability and selectivity…
- Line 62: mining?
- What was the moisture content in the sample? How long was it stored at 0°C? Did any chemical and microbiological changes occur during storage of the water content is high?
- Methodology for the conventional extraction with water is missing in materials and methods.
- Lines 379-381: How did you choose temperature and extraction time? Is the sample to solvent ratio too low? How did you choose that?
- Lines 148-152: Values in text are not in accordance with the values from Table 1. Please check and correct the discussion.
- Line 242: The order of betacyanin content is not in accordance with the Table 2. Please check.
- Methodology for the betacyanin and betaxanthin content is missing in materials and methods. Please add.
- Lines 315-324: Please move to materials and methods.
- Potential application, challenges and limitations of these extracts should be highlighted in the Conclusions.
Author Response
REVIEW 1
Present research by Hernandez-Aguirre et al. investigates extraction and stabilization of beetroot pigments. Deep eutectic solvents (DES) based on urea and magnesium chloride hexahydrate were synthetized, characterized and applied as extraction solvents. Betalain pigment yield was compared with traditional extraction with water and stability study was performed on obtained extracts. Extracts obtained by DES provided clear advantages comparing to conventional extracts and present study should be very interesting since it provides interesting concept of waste valorization. Following remarks and questions must be accessed by the authors in order to improve the paper prior the further reconsideration.
Lines 30-32: Current application of beetroot was should be given.
Response: Bioethanol process was also included as current application of beetroot.
Lines 53-55: Please brief explanations of these references instead of just listing the authors.
Response: We explained the [13] reference because the principal input of [11, 12] is used below.
Line 60: the stability and selectivity…
Response: the indicated text was corrected (line 66)
Line 62: mining?
Response: the mining word was corrected by extracting (line 68)
What was the moisture content in the sample? How long was it stored at 0°C? Did any chemical and microbiological changes occur during storage of the water content is high?
Response: The samples were stored for a maximum of 3 weeks. There were no chemical and microbiological changes. The water content in samples of beetroot wastes and samples of defrost wastes were included in methodology, lines 359-364.
Methodology for the conventional extraction with water is missing in materials and methods.
Lines 379-381: How did you choose temperature and extraction time? Is the sample to solvent ratio too low? How did you choose that?
Response: The sample to solvent ratio was corrected in the text (line 392); the correct mass of sample is 0.5 g, obtaining a solid-to-liquid ratio of 1:30 g/mL (this data was included in line 393). Extraction temperature and time (25°C and 900 s) are relative to vortex agitation. However previous sonication for 3h was applied in extraction process.
Lines 148-152: Values in text are not in accordance with the values from Table 1. Please check and correct the discussion.
Response: Values were corrected. The temperature values of Table 1 were standardized as T°C
Line 242: The order of betacyanin content is not in accordance with the Table 2. Please check.
Response: Order of betalain content in BEW data of Table 2 were corrected.
Methodology for the betacyanin and betaxanthin content is missing in materials and methods. Please add.
Response: Determination content of betacyanin and betaxanthin is expressed in lines 398-407
Lines 315-324: Please move to materials and methods.
Response: Lines 315-324 are results of betalains degradation. The text was marked in line 325-328.
Potential application, challenges and limitations of these extracts should be highlighted in the Conclusions.
Response: Lines 433-444 highlight the potential application and limitations of research.

Reviewer 2 Report
The application of DES as alternatives to organic solvents is interesting and work done with these is always of interest.
However considering that the title of the paper refers to "Re-use", the authors should provide realistic examples of how this particular DES extract containing urea and Mg chloride can be used, considering its very unpleasant organoleptic properties. Otherwise, it is just an example of stability enhancing properties of this DES and I suggest choosing a more fitting title.
Please take note of the following:
Line 48 - using DES is not an auxiliary technique, it is an alternative extraction solvent, please reword.
Line 57 and reference to NADES: this is not accurate, NADES can be donors, acceptors or neither; actually it is not clear why it is necessary to mention them at all, but if done, please do so correctly.
There are terms that should be corrected (apart from the English):
-change all references to "chloride of magnesium" for the conventional nomenclature in English, i.e., magnesium chloride.
- what is meant by the unit mL.mol-cm? Please correct.
Regarding the method, please clarify:
- (a) how many replicates were prepared for each extraction condition?
- (b) you refer to efficiency of extraction between the tested DES and claim the method to be efficient. Please describe what alternatives you tested on these samples to support your claim.
- (c) clarify what work was done to optimise extraction conditions, for example why you choose 900 s as an extraction time.
- Please have your manuscript re-edited to correct both grammar and use of English errors.
Author Response
REVIEW 2
The application of DES as alternatives to organic solvents is interesting and work done with these is always of interest.
However considering that the title of the paper refers to "Re-use", the authors should provide realistic examples of how this particular DES extract containing urea and Mg chloride can be used, considering its very unpleasant organoleptic properties. Otherwise, it is just an example of stability enhancing properties of this DES and I suggest choosing a more fitting title.
Response: Authors consider that the title of manuscript is correct “Extraction and stabilization of betalains from beetroot wastes using Deep Eutectic Solvents” because this proposal exposes a extraction methods of betalains. As DES extracts, they have limitations in food area, but they could be used for other functions. Limitations and potential use were indicated in lines 433-444.
Please take note of the following:
Line 48 - using DES is not an auxiliary technique, it is an alternative extraction solvent, please reword.
Response: The text was corrected. This can be seen in line 50 of the new version of manuscript.
Line 57 and reference to NADES: this is not accurate, NADES can be donors, acceptors or neither; actually it is not clear why it is necessary to mention them at all, but if done, please do so correctly.
Response: The text was corrected. This can be seen in line 69-71 of the new version of manuscript.
There are terms that should be corrected (apart from the English):
-change all references to "chloride of magnesium" for the conventional nomenclature in English, i.e., magnesium chloride.
Response: chloride of magnesium was changed by magnesium chloride
- what is meant by the unit mL.mol-cm? Please correct.
Response: Units of extinction molar coefficient were corrected. This can be seen in line 406-407 of the new version of manuscript
Regarding the method, please clarify:
(a) how many replicates were prepared for each extraction condition?
Response: three replicates
(b) you refer to efficiency of extraction between the tested DES and claim the method to be efficient. Please describe what alternatives you tested on these samples to support your claim.
Response: The alternatives used to extract betalains were by polar solvents and acid pH
(c) clarify what work was done to optimise extraction conditions, for example why you choose 900 s as an extraction time.
Response: We have corrected and complemented the text of method in the new version of manuscript. For betalains extraction, we used a blender, an ultrasonic bath for 3h, followed by a vortex agitation for 900 s. The betalins extraction time in ultrasonic bath was previously studied for 3, 4 and 5 h, establishing 3 h as optimal condition, in this step because after we finished the extraction by a vortex agitation.
- Please have your manuscript re-edited to correct both grammar and use of English errors.
Response: We review the manuscript for detected English errors.

Reviewer 3 Report
The research article gives a scientific value on extracting betalains from beetroot wastes. However there are few issues that need to be clarified for better comprehension.
Comments are listed below:
- Please provide the scientific name of the beetroot used both in title and the material section. It is important to know the species of the beetroot used because the betalain contents might vary depending on different beetroot.
- Overall the manuscript is well written but the reference style is somehow a bit incorrect. Please refer to MDPI’s reference style. The last name is out front instead of the abbreviated first name.
- In Table 2, how the yield was calculated? The authors should provide an explanation of the equation in footnote or in the material and method section.
- What was the real yield of betalins out of the fresh beetroot? Can the authors provide the data because it is crucial to the production cost especially when the production of betalins in larger scale? Did the authors conduct the proximate analysis of beetroot wastes?
- How is the extracted betalains being used in food application? Does the compounded solvent need to be removed prior to further application? If so, please address it in conclusion.
- It would be nice to see the chemical structures of betacyanin, betaxanthin and betalain alone with the extracting solvents. Can the authors provide a figure illustrating the possible mechanism or the process flow?
Author Response
REVIEW 3
The research article gives a scientific value on extracting betalains from beetroot wastes. However there are few issues that need to be clarified for better comprehension.
Comments are listed below:
- Please provide the scientific name of the beetroot used both in title and the material section. It is important to know the species of the beetroot used because the betalain contents might vary depending on different beetroot.
Response: Scientific name of beetroot was included in the title and abstract of the manuscript (line 10). Wastes are from red beetroot (red cloud variety).
Overall the manuscript is well written but the reference style is somehow a bit incorrect. Please refer to MDPI’s reference style. The last name is out front instead of the abbreviated first name.
Response: Authors checked and corrected the references in all manuscript
- In Table 2, how the yield was calculated? The authors should provide an explanation of the equation in footnote or in the material and method section.
Response: The yield in the extraction of betalains was determined taking as a reference betalains content in whole beetroot, which was determined as 4. 67 mg/g. Equation was added in methodology section, lines 416-419.
- What was the real yield of betalains out of the fresh beetroot? Can the authors provide the data because it is crucial to the production cost especially when the production of betalins in larger scale? Did the authors conduct the proximate analysis of beetroot wastes?
Response: Authors determined water and betalains content. Data were included in methodology lines 359-364. The yield in the extraction of betalains was determined taking as a reference betalains content in whole beetroot, which was determined as 4. 67 mg/g. Equation was added in methodology section, lines 416-419. Also, the betalains yield can be seen in betalains content (mg)/g of beetroot wastes.
- How is the extracted betalains being used in food application? Does the compounded solvent need to be removed prior to further application? If so, please address it in conclusion.
Response: Possible application and required research was included in conclusions (Line 433-444).
- It would be nice to see the chemical structures of betacyanin, betaxanthin and betalain alone with the extracting solvents. Can the authors provide a figure illustrating the possible mechanism or the process flow?
Response: DES was used as solvent agent to betalains extraction. Therefore, the mechanism is relative to dissolution process, which occurs by separation the elemental entities that make up the solute and the solvent separately, overcoming the solute-solute, solvent-solvent and after solute-solvent interactions by attraction of elemental entities of the solute and the solvent (solvation). However, the description of separation the elemental entities of betalains and DES is complex.

Round 2
Reviewer 1 Report
Authors responded on all my remarks. I don't have additional comments.
Author Response
English language and style are fine/minor spell check required
Response: The paper was review by a native English speaker
Reviewer 2 Report
Dear authors,
The manuscript is now clearer regarding some of the previous issues.
However, there are still some parts that require clarification:
Line 415: "The extinction coefficients for betacyanin 6 000 000 M-1.cm-1 and 48 000 000 M-1.cm-1 for betaxanthin are relative to ε. A conversion factor of 1000 was used to change 416 g to mg. "
The literature value is 60000 M-1cm-1 for betacyanin and 48000M-1cm-1 ; could you provide references for those numbers you have used? Or is your unit mM? Your conversion should not be done by changing the extinction coefficient value!
- Formatting:
Please use correct chemical notation throughout: subindexes for MgCl2 and H2O for example in captions of figures; or superindexes when required, e.g. M-1 in line 415.
English is improved but still requires correction (some suggestions to improve confusing use of English, other remaining errors are style-related)
- line 43: please change the word "compound" as a descriptor of DES/NADES since they are mixtures of compounds, perhaps "substances" is at least less incorrect.
- line 330: please change "whose solution,etc" for the simpler "as expressed in Eq....
- line 463: not sure what you mean but perhaps "beetroot wastes included peels and pulp" for example; but whatever, please correct the phrase as they are not "relatives".
In general, the paper would benefit from the revision by a native English speaker.
Author Response
REVIEW-2
The manuscript is now clearer regarding some of the previous issues.
However, there are still some parts that require clarification:
Line 415: "The extinction coefficients for betacyanin 6 000 000 M-1.cm-1 and 48 000 000 M-1.cm-1 for betaxanthin are relative to ε. A conversion factor of 1000 was used to change 416 g to mg. "The literature value is 60000 M-1cm-1 for betacyanin and 48000M-1cm-1 ; could you provide references for those numbers you have used? Or is your unit mM? Your conversion should not be done by changing the extinction coefficient value!
Response: Values of extinction coefficient were corrected; the correct values are 60000 M-1 cm-1 for betacyanin and 48000 M1 cm-1 . The corrections was marked with red text. Line 415.
Formatting:
Please use correct chemical notation throughout: subindexes for MgCl2 and H2O for example in captions of figures; or superindexes when required, e.g. M-1 in line 415.
Response: All the writing was revised to make the indicated corrections. Titles of Figure 1 and 2 were corrected and line 415.
English is improved, but still requires correction (some suggestions to improve confusing use of English, other remaining errors are style-related)
line 43: please change the word "compound" as a descriptor of DES/NADES since they are mixtures of compounds, perhaps "substances" is at least less incorrect.
Response: According to observation, Line 50, 71 were corrected.
line 330: please change "whose solution, etc" for the simpler "as expressed in Eq....
Response: According to observation, the text was changed. Line 331
line 463: not sure what you mean but perhaps "beetroot wastes included peels and pulp" for example; but whatever, please correct the phrase as they are not "relatives".
Response: According to observation, the text was changed. Line 364
In general, the paper would benefit from the revision by a native English speaker.
Response: The paper was review by a native English speaker